# Learning Distinct and Representative Modes for Image Captioning

**Qi Chen**[*]    **Chaorui Deng**[*]    **Qi Wu**[†]
Australian Institute for Machine Learning, University of Adelaide
`{qi.chen04, chaorui.deng, qi.wu01}@adelaide.edu.au`

## Abstract

Over the years, state-of-the-art (SoTA) image captioning methods have achieved promising results on some evaluation metrics (*e.g.*, CIDEr). However, recent findings show that the captions generated by these methods tend to be biased toward the "average" caption that only captures the most general mode (*a.k.a*, language pattern) in the training corpus, *i.e.*, the so-called *mode collapse* problem. Affected by it, the generated captions are limited in diversity and usually less informative than natural image descriptions made by humans. In this paper, we seek to avoid this problem by proposing a Discrete Mode Learning (DML) paradigm for image captioning. Our innovative idea is to explore the rich modes in the training caption corpus to learn a set of "mode embeddings", and further use them to control the mode of the generated captions for existing image captioning models. Specifically, the proposed DML optimizes a dual architecture that consists of an image-conditioned discrete variational autoencoder (CdVAE) branch and a mode-conditioned image captioning (MIC) branch. The CdVAE branch maps each image caption to one of the mode embeddings stored in a learned codebook, and is trained with a pure non-autoregressive generation objective to make the modes distinct and representative. The MIC branch can be simply modified from an existing image captioning model, where the mode embedding is added to the original word embeddings as the control signal. In the experiments, we apply the proposed DML to two widely used image captioning models, Transformer and AoANet. The results show that the learned mode embedding successfully facilitates these models to generate high-quality image captions with different modes, further leading to better performance for both diversity and quality on the MSCOCO dataset[1].

## 1 Introduction

Image captioning aims to generate natural descriptions for a given image. It is widely used in many real-world applications such as human-computer interaction, multi-modal recommendation, and hence has attracted lots of research attention. Recently, many state-of-the-art (SoTA) methods [20, 26, 57] have achieved promising results *w.r.t.* evaluation metrics like CIDEr [47], BLEU [38], and SPICE [1]. However, as discussed in [52], focusing on achieving higher scores on these metrics usually biases the image captioning models towards using only the common words, phrases and language patterns in the training corpus when describing the images (see Figure 1 for an example). In other words, the model automatically finds the most general mode to perform captioning. As a result, the generated captions are limited in diversity both semantically and syntactically. This is far away from the ability of human beings as humans are able to describe the image in various ways.

---

[1]Code is available at `https://github.com/bladewaltz1/ModeCap`
[*]Authors contributed equally.    [†]Corresponding author.

36th Conference on Neural Information Processing Systems (NeurIPS 2022).

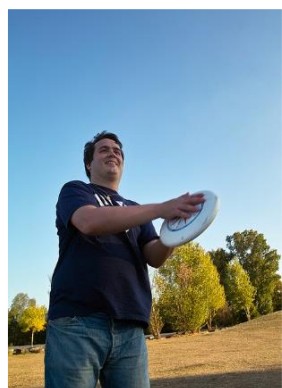

**Captions generated by humans:**
*A man holding a white frisbee while standing on a field.*
*A man standing outside holding a frisbee in his hands.*
*A man is outside holding onto a frisbee disc.*
*A smiling man in the park with a frisbee.*
*A man is having a good time playing frisbee.*

**Captions generated by existing models:**
Transformer: *A man holding a frisbee while standing in a field.*
AoANet: *A man holding a frisbee in a field.*

**Captions generated by DML (Ours):**
Mode-7:   *There is a man playing with a frisbee.*
Mode-32: *Man in a blue shirt throwing a white frisbee.*
Mode-43: *A close up of a person with a frisbee.*
Mode-3:   *A young man holding a white frisbee in his right hand.*
Mode-58: *A man is about to throw a frisbee.*

Figure 1: Captions written by humans vs. captions generated by existing models (Transformer [46] and AoANet [20]), and by our DML. The model-generated captions prefer to use common words or phrases while captions derived from humans are more informative and diverse. The proposed DML generates diverse image captions based on different mode embeddings, and some of the modes tend to yield certain language patterns. *E.g.*, mode-7 is likely to generate captions with the pattern of *"There is ..."*, mode-3 tends to produce complex sentences while mode-58 is prone to brief sentences.

The cause of this phenomenon is widely known as the *mode collapse* problem and has been discussed in many prior works on generative modeling [17, 56]. Formally, given an input $x$ and a generative model $\mathcal{G}$, mode collapse appears when the estimated output distribution $P_{\mathcal{G}}(x)$ assigns most of its probability mass to a small region of the output space, despite that the real data distribution $P_{\text{data}}(y)$ has a much larger variance. As a consequence, the randomly sampled outputs $\{y_i \sim P_{\mathcal{G}}(x) \mid i \in \mathbb{N}\}$ tend to be very similar. While mode collapse is typically a side effect for generative modeling, it is somewhat "welcomed" in SoTA image captioning models as it usually facilitates a higher evaluation performance on reference-based metrics like CIDEr, BLEU and SPICE. For example, CIDEr optimization [41] based methods [11, 20] have significantly pushed the performance of image captioning to a new level on mainstream reference-based evaluation metrics. However, as shown in [31, 52], the success of CIDEr optimization could be largely attributed to its ability of reducing the modes in the generated captions.

Some recent researches in image captioning have attempted to tackle the mode collapse problem so as to improve the diversity of the generated captions. Specifically, [3, 33, 50] adopt conditional variational autoencoders (CVAE) to encode the conditional distribution of the image captions into a low-dimensional continuous latent space $\mathcal{Z}$. When performing inference, these methods randomly sample latent variables from $\mathcal{Z}$ and input them into the caption decoder to drive it towards using different language patterns to describe the image. In this sense, the sampled latent variables can be considered as the continuous representations of modes interpolated from all occurred modes during training, thus alleviating the mode collapse phenomena to some extent. Nevertheless, it is still difficult to interpret the underlying conditional distribution, *i.e.*, which part of $\mathcal{Z}$ corresponds to which kind of language patterns in the real world. Thus, for the captions derived from two sampled mode representations, it is uncertain how they differ from each other during the sampling process, as well as their qualities. To tackle this uncertainty, another line of works seeks to learn controllable image captioning models. For example, [9, 44] control the style of the image captions with learned sentiment or personality representations like "factual" or "humorous", "positive" or "negative", *etc*. [15] controls the syntactic structure of the image captions through part-of-speech tags. [10] focuses on different image regions to generate region-specific image captions. However, these methods rely on additional tools or annotations to supervise the learning of modes. More critically, this also restricts their modes within a pre-defined domain.

In this paper, we tackle the above problems by learning distinct and representative modes for image captioning through a new Discrete Mode Learning (DML) paradigm. Different from CVAE-based diverse image captioning methods, DML learns a codebook, which is an embedding matrix consisting of a set of mode embeddings that spans a discrete latent space, thus the language patterns encoded in different modes can be easily evaluated and are more perceptible. Different from previous controllable

image captioning methods, DML requires no additional supervision for mode and hence is more convenient to use and is not limited to the pre-defined set of control signals.

Specifically, we optimize a dual architecture in the proposed DML: **Firstly**, an image-conditioned discrete variational autoencoder (CdVAE) branch, where an encoder extracts the hidden states for all the reference captions paired with an image, and further quantizes them using their matched mode embeddings in the codebook according to Euclidean distance. For the matching algorithm, we choose Hungarian algorithm instead of the naive nearest neighbor look-up, which we find to be critical for increasing the number of effective modes. Afterward, a decoder is used to reconstruct the reference captions in a *fully non-autoregressive* manner, which breaks the sequential dependencies on previous tokens and enforces the decoder to rely purely on the mode embeddings to generate different captions based on the same image feature, leading to more distinct and representative mode embeddings. **Secondly**, a mode-conditioned image captioning (MIC) branch, which can be simply modified from an existing image captioning model by adding the mode embedding to the original word embeddings as a control signal. During inference, the CdVAE branch is dropped, and the MIC branch is used to generate image captions with various language patterns according to the mode embeddings in the codebook.

In the experiments, we evaluate the effectiveness of our proposed DML paradigm by applying it to the widely-used Transformer [46] and the state-of-the-art AoANet [20], denoted by Transformer-DML and AoANet-DML, respectively. We observe that some of the modes tend to yield clear language patterns in the generated captions, as shown in Figure 1, demonstrating that DML has successfully learned perceptible modes without explicit supervision. We also find that our models perform surprisingly well under diversity evaluation (using metrics like SelfCIDEr [52]) and oracle performance evaluation (on mainstream reference-based metrics like CIDEr [47]), achieving new state-of-the-art results. This shows that our learned modes are not only distinct but also effectively cover the rich modes that appeared within the dataset, which suggests that they are very representative. Moreover, we find that the models trained with DML outperform their original counterparts in terms of quality on some of the modes, meaning that DML can serve as a cost-free plugin for existing image captioning models.

## 2 Related Works

**Image captioning.** Image captioning [2, 20, 49, 53] seeks to generate descriptions based on the given images, which has received lots of attention from the researchers [23]. The conventional paradigm of image captioning models [16, 21, 49] mainly consists of two parts: a CNN-based image encoder and an RNN-based decoder for caption generation. Based on this diagram, many works [2, 20, 57] introduce the attention mechanism [42, 46, 14, 25, 53], which enforces models to consider more about the highlighted regions. Besides, to improve the performance, [54, 55] explore the visual relationships by constructing a semantic or scene graph, while [18, 41] optimize their models by Reinforcement Learning (RL) and directly use CIDEr [47] to compute the reward. However, these image captioning models mainly consider how to achieve higher evaluation scores, which usually bias the generated captions to an "average" version that contains the common words and phrases in the training corpus only.

**Diverse and controllable image captioning.** Diverse image captioning aims at learning a model that can generate various captions based on the same image. To this end, CVAE-based models [6, 33, 50] learn a latent space during training and then generate diverse captions by sampling different priors from the latent space. GAN-based models [12, 24, 43] predict diverse captions by using different random noises as inputs accompanied with the given images. Although the diverse image captioning models are able to produce different captions during inference, it is still non-trivial to interpret the underlying conditional distribution, leading to an uncertainty of the model behavior.

To make the generated captions controllable, some works [7, 8, 10, 13, 35, 36] introduce an additional control signal. For example, Mathews *et al.* [35] provide a sentiment signal for each caption, and seek to control the sentiment of the generated captions. Deng & Ding [13] take the length of the caption as a control signal. Conditioned on different length level embeddings, the model is able to generate length-controllable descriptions for the input image. However, most of these methods have to rely on additional tools or annotations to supervise the learning of the control signal, which is usually inconvenient to collect and further limits the language pattern within a pre-defined domain.

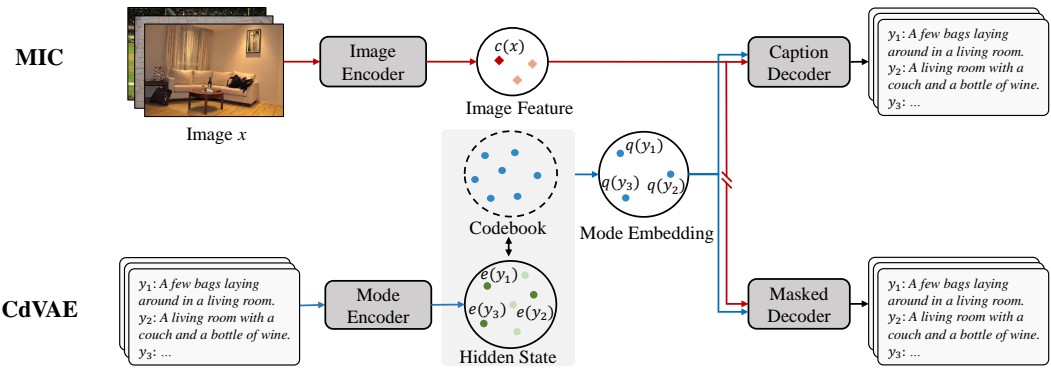

Figure 2: Overall of the DML paradigm. The "$\leftrightarrow$" denotes the matching operation by using Hungarian algorithm while the "\\" means there is no gradient flow.

## 3 Method

The overall architecture of the proposed Discrete Mode Learning (DML) paradigm is illustrated in Figure 2. It consists of two branches: an image-conditioned discrete variational autoencoder (CdVAE) branch, which contains a mode encoder $\mathcal{E}_m$ and a masked decoder $\mathcal{D}_m$; and a mode-conditioned image captioning (MIC) branch consists of an image encoder $\mathcal{E}_c$ and a caption decoder $\mathcal{D}_c$. The two branches are connected through a codebook $\Omega$ that contains a set of mode embeddings learned through the DML paradigm. During training, the model takes as inputs an image $x$ and its paired reference captions $\{y_i\}_{i=1}^n$, and no additional supervision is required. During inference, the CdVAE branch is dropped, and the MIC branch is used to generate image captions with various language patterns according to the mode embeddings in the codebook. More details are as follows.

### 3.1 Discrete Mode Learning

In a typical image captioning model, the training objective is to maximize $\log p(y_i|x)$ for each $y_i$ in $\{y_i\}_{i=1}^n$, which is ill-posed since the model is optimized to approach multiple different targets conditioned on the same input $x$, and usually leads to a mode collapse problem. Previous works alleviate this problem by introducing a latent variable $z_i$ to the objective function, *i.e.*, $\mathbb{E}_{z_i \sim p(z)}[\log p(y_i|x, z_i)]$, which serves as an explicit indication of mode and drives the model to generate different targets conditioning on different $z_i$. *E.g.*, in CVAE-based diverse image captioning models [33, 50], $z_i$ is randomly sampled from a continuous latent space, which leads to an uncertainty of the model behavior. While in some controllable image captioning models [10, 15], $z_i$ needs to be pre-defined with the help of additional tools or annotations, which is usually restricted to a small set.

Unlike these methods, our DML samples latent variables from an embedding matrix $\Omega \in \mathbb{R}^{k \times d}$ which we call codebook. It defines a discrete latent space, where each entry in $\Omega$ corresponds to a potential mode embedding, and $k$ is a hyper-parameter representing the total number of modes in the codebook. During training, the mode encoder $\mathcal{E}_m$ is used to extract the representation for each caption $y_i$, denoted by $e(y_i)$, and match the representation with one of the entries in $\Omega$. The matched entry is then adopted as the mode embedding of $y_i$, denoted by $q(y_i)$, and the objective becomes:

$$\max_{\mathcal{E}_m, \mathcal{D}_m, \Omega} \log p(y_i|q(y_i), x), \quad \text{where } q(y_i) = \texttt{Match}(e(y_i), \Omega). \tag{1}$$

Here, $p(y_i|q(y_i), x)$ is the conditional distribution of the caption $y_i$ given its mode embedding $q(y_i)$ and the paired image $x$, which is approximated with the masked decoder $\mathcal{D}_m$. $\texttt{Match}(\cdot, \cdot)$ indicates the matching operation. Note that, there is usually no real gradient defined for typical matching operators like the nearest neighbor look-up. Therefore, we follow [45] to first use the straight-through estimator to directly copy the gradients from $q(y_i)$ to $e(y_i)$, and then apply Vector Quantization algorithm to move $q(y_i)$ towards $e(y_i)$ through mean square loss so as to train the matched codebook entries. The new objective function is

$$-\log p(y_i|q(y_i), x) + \|\text{sg}[e(y_i)] - q(y_i)\|_2^2 + \beta\|e(y_i) - \text{sg}[q(y_i)]\|_2^2, \tag{2}$$

where the first term inherited from Eq. (1) is used to optimize $\mathcal{E}_m$ and $\mathcal{D}_m$. The second term is used to train the matched entries in $\Omega$. The last term is a commitment loss that is responsible for bounding the embedding space of $\Omega$. $\text{sg}[\cdot]$ refers to "stop gradient" and $\beta$ is a hyper-parameter.

## 3.2 Mode Encoding and Assignment

We adopt a stack of $N_e$ transformer encoder layers as our mode encoder $\mathcal{E}_m$. The input of $\mathcal{E}_m$ is $y_i$ appended with a special [MODE] token to the start. The output of $\mathcal{E}_m$, $e(y_i) \in \mathbb{R}^d$, is the hidden state of the [MODE] token from the last layer. When performing the matching process in Eq. (1), a naive approach is to use the nearest neighbor look-up algorithm, *i.e.*,

$$q(y_i) = \Omega_\iota, \quad \text{where } \iota = \arg\min_j \|e(y_i) - \Omega_j\|_2. \tag{3}$$

However, we find in the experiments that in the models trained with this assignment strategy, the output of $\mathcal{E}_m$ quickly converges to the nearby of two or three mode embeddings in $\Omega$, which is a clear sign of mode collapse (see Figure 5).

To avoid this, inspired by the object query assignment in [5], we treat the mode assignment of the captions as a Bipartite Graph Matching problem and solve it using Hungarian algorithm. Specifically, for all the reference captions $\{y_i\}_{i=1}^n$ paired with an image, we construct a bipartite graph between the output hidden states $\{e(y_i)\}_{i=1}^n$ from $\mathcal{E}_m$ and the mode embeddings in $\Omega$. We first pad $\{e_i\}_{i=1}^n$ to the size of the codebook, $k$, with $\varnothing$ (assume $n < k$). Then, we search for a permutation of $k$ elements $\tau \in \mathfrak{S}_k$ with the lowest assignment cost:

$$\hat{\tau} = \arg\min_{\tau \in \mathfrak{S}_k} \sum_i^k \|e(y_i) - \Omega_{\tau(i)}\|_2, \tag{4}$$

and the assigned mode embedding for each $y_i$ is $q(y_i) = \Omega_{\hat{\tau}(i)}$. We find this assignment strategy greatly increases the number of effective mode embeddings in $\Omega$ (see Figure 5).

## 3.3 Fully Non-autoregressive Decoder

After getting the mode embeddings $q(y_i)$ for each caption $y_i$, we feed it together with the image features from $x$ into a decoder to generate $y_i$ and estimate the conditional distribution $p(y_i|q(y_i), x)$ in Eq. (2). In general, sequence generation models are often trained through the autoregressive Teacher Forcing scheme, which aims to maximize the likelihood of the ground-truth token $w_t$ given all preceding ground-truth tokens $w_{j<t}$. In our setting, the objective function would be:

$$-\log p(y_i|q(y_i), x) = \sum_{t=1}^T -\log p(w_t|w_{j<t}, q(y_i), x) \tag{5}$$

where $w_t$ is the $t$-th token in $y_i$ and $T$ is the total number of tokens in $y_i$. However, being able to access the previous tokens causes the decoder to ignore the mode embedding when reconstructing $y_i$, since the previous tokens already provide enough information for the mode of $y_i$. As a result, the training signals for the mode embeddings could be useless.

Therefore, we propose to use a fully non-autoregressive (NAT) objective for the CdVAE branch of DML to facilitate the training of the mode embeddings. Specifically, the masked decoder $\mathcal{D}_m$ is a stack of $N_d$ transformer decoder layers that takes $q(y_i)$, $x$, and a sequence of $T$ [MASK] tokens as input, and predicts each target token in $y_i$ in a conditionally-independent manner, *i.e.*,

$$-\log p(y_i|q(y_i), x) = \sum_{t=1}^T -\log p(w_t|\text{[MASK]}, q(y_i), x). \tag{6}$$

Apply this equation to Eq. (2) will result in our final objective function for the CdVAE branch. We find this leads to more distinct and representative mode embeddings (see Figure 5).

## 3.4 Learning MIC Models with DML

So far we have introduced how DML learns the mode embeddings with the CdVAE branch. Here we show how to attach the CdVAE branch to an existing image captioning model to make it aware of the

mode of the generated captions. Specifically, in a standard encoder-decoder-based image captioning model, the encoder $\mathcal{E}_c$ encodes the information from the image $x$ into a sequence of hidden states, then the decoder $\mathcal{D}_c$ attends to the encoder hidden states and predicts the ground-truth tokens in the reference caption $y_i$ following the Teach Forcing scheme similar to Eq. (5).

To adapt it into our DML framework, we only need to make a minor modification to the token embedding layer of $\mathcal{D}_c$ by adding the mode embedding $q(y_i)$ to the word embeddings of each token in $y_i$ (*i.e.*, $w_{j<t}$ in Eq. (5)) element-wisely, resulting in our MIC branch. The training of MIC and CdVAE is performed jointly, using the objective functions in Eq. (2) and Eq. (5), respectively. In practice, we find the CdVAE branch is much harder to train than the MIC branch, due to its non-autoregressive prediction manner. To make the convergence speed of the two branches compatible, we let them use different batch sizes, *i.e.*, for each image, the CdVAE branch takes as input all its paired captions $\{y_i\}_{i=1}^n$, while the MIC branch randomly samples just one caption from $\{y_i\}_{i=1}^n$. Moreover, the gradient from the CdVAE branch will not be back-propagated to the image encoder.

**Inference.** During inference, the CdVAE branch is dropped, and the inference of MIC follows a very similar procedure as the original image captioning model. The only difference is that it requires selecting a mode embedding from the codebook and adding it to the input token embeddings.

# 4   Experiment

## 4.1   Dataset and Evaluation Metrics

**Dataset.** We train and evaluate our method on MSCOCO dataset [28] that contains $123,287$ images and each image is corresponding to at least 5 captions. For a fair comparison, we follow the previous works [32, 33] in the area of diverse and controllable image captioning to use the m-RNN split [34] of the COCO dataset, which divides the data into $118,287$, $4,000$ and $1,000$ for training, validation and testing, respectively.

**Quality evaluation.** To assess the quality of the generated captions, we use five widely used evaluation metrics, *i.e.*, BLEU [38], ROUGE [27], METEOR [4], CIDEr [47], and SPICE [1]. Moreover, to further evaluate the performance of the generated captions, we employ another CLIP-based [39] metric namely ClipScore [19], which can assess whether the generated captions are semantically aligned with given images, even when they are totally different from the reference captions.

**Diversity evaluation.** To investigate the diversity of the generated captions, we use SelfCIDEr [51], mBLEU, and n-gram diversity (*i.e.*, Div-$n$ [3]). All of these metrics evaluate the diversity by comparing the $n$-gram differences among the generated captions that belong to the same image.

## 4.2   Implementation Details

**Choice of base models.** The proposed DML is a general learning paradigm and we expect it can be easily applied in many existing image captioning models and improve their controllability and diversity. Thus, for the MIC branch, we choose two widely-used and representative architectures as our base models, *i.e.*, Transformer [46] and AoANet [20], denoted by Transformer-DML and AoANet-DML separately, to show the generalization ability of DML. Most current SoTA image captioning models, like M2Transformer [11], XLAN [37] and many vision-language pre-training models, are based on the Transformer architecture. Moreover, the performance of Transformer and AoANet is also competitive with the SoTA models [30] when vision-language pre-training is not performed. Therefore, they are good baseline models to illustrate the performance of our DML paradigm.

**Detailed settings.** In the CdVAE branch, the number of transformer layers for $\mathcal{E}_m$ and $\mathcal{D}_m$ is set to 6 and 2, respectively. We use 12 attention heads and a hidden size of 768 for all transformer layers. $\beta$ in Eq. (2) is set to 0.25, following [45]. The number of mode embeddings in $\Omega$ is set to 64 by default. We convert each image to 100 object proposals by Faster RCNN [40] pre-trained on Visual Genome [22]. We train the models for 100,000 iterations with a batch of 64 images and all the paired captions. We use AdamW [29] optimizer with a learning rate of 2e-4, and cosine decay it to 0. We use the label smoothing of 0.1 and the gradient clipping threshold of 1.0. We train Transformer-DML and AoANet-DML on one NVIDIA 3090 GPU with about 10 and 13 GPU hours, respectively.

Table 1: Diversity evaluation on the best-5 sentences obtained from consensus re-ranking.

| Methods | LNFMM [32] | COS-CVAE [33] | Seq-CVAE [3] | Transformer-BS | Transformer-DML |
|---|---|---|---|---|---|
| Div-1 (↑) | 0.37 | 0.39 | 0.33 | 0.21 | **0.43** |
| Div-2 (↑) | 0.50 | 0.57 | 0.48 | 0.29 | **0.59** |
| SelfCIDEr (↑) | - | 0.79 | - | 0.57 | **0.83** |
| mBLEU (↓) | 0.64 | **0.53** | 0.64 | 0.78 | 0.54 |

| | Image | | |
|---|---|---|---|
| Image |  |  |  |
| Mode 7 | *There is a man in the middle of a field playing with a frisbee* | *There is a man riding a motorcycle down the street* | *There is a man on a skateboard holding a can* |
| Mode 32 | *Man in red shirt throwing a white frisbee* | *Man on a yellow motorcycle driving down the road* | *Man on skateboard with a beer on the ground* |
| Mode 43 | *A close up of a person with a frisbee* | *A close up of a person riding a motorcycle on a road* | *A close up of a person on a skate board* |
| Mode 3 | *A young man holding a white frisbee on top of a green field* | *A man riding a yellow motorcycle with a yellow helmet on* | *A young man holding a bottle of water while standing on a skateboard* |
| Mode 58 | *A man getting ready to throw a frisbee* | *A man that is sitting on a motorcycle* | *A person on a court with a skateboard* |

Figure 3: Samples of captions that are generated based on different modes. Mode-7, mode-32 and mode-43 tend to generate the captions with a certain pattern, *i.e.*, *"There is ..."*, *"Man ..."*, and *"A close up of ..."*. Mode-3 is apt to use long sentences while mode-58 is prone to brief sentences.

## 4.3 Evaluation on Diversity

**Quantitative results.** In this part, we perform a diversity analysis for the proposed DML paradigm based on Transformer-DML. We compare our model with previous SoTA methods as well as Beam Search (BS) and show the results in Table 1. From the table, our DML-based model achieves higher performance on most of the diversity evaluation metrics, which demonstrates its effectiveness in learning distinct modes.

**Qualitative results.** We further provide several samples of the generated captions of our Transformer-DML model in Figure 3. We find that the learned modes exhibit clear and distinct language patterns. More specifically, mode-7, mode-32 and mode-43 tend to follow the patterns *"There is ..."*, *"Man ..."*, and *"A close up of ..."*, respectively. Mode-3 and mode-58 focus more on the semantic complexity of the caption, *i.e.*, mode-58 is likely to generate a brief caption while mode-3 is prone to the complicated one. These results show that for the same image, the mode embeddings learned through DML facilitate the model to generate captions in various and comprehensible ways.

## 4.4 Evaluation on Quality

**Oracle results.** To investigate whether the embeddings in the codebook are able to cover the rich modes that appeared within the dataset, we calculate the caption evaluation metrics in the oracle setting consistent with prior works [3, 32, 33], *i.e.*, taking the maximum score for each quality metric over all the candidate captions for each image. Specifically, we train Transformer-DML and AoANet-DML with codebook sizes $k = 20$ and $k = 100$, and evaluate the oracle results of the captions generated by all modes. In Table 2, the models with DML obtain the best and second best results on all the evaluation metrics *w.r.t.* both 20 and 100 samples. Moreover, we compare our DML paradigm with the SoTA baseline COS-CVAE by calculating the oracle scores of CIDEr, SPICE and METEOR with different numbers of samples. In Figure 4, the proposed DML consistently outperforms COS-CVAE on all settings. The high gain in quality metrics demonstrates that the proposed DML successfully captures the rich and representative modes in the training corpus.

**Results of individual mode embedding.** To assess the quality of captions generated using different modes, we calculate the evaluation scores of captions generated from some representative modes that are manually selected with distinct patterns. In Table 3, on mode-58, Transformer-DML outperforms the original Transformer on all reference-based metrics. Moreover, the Transformer-DML with different mode embeddings can achieve better or at least competitive performance compared with the

Table 2: Comparison with baselines *w.r.t.* oracle performance (*i.e.*, best-1 quality) on COCO dataset. "#Sample" refers to the number of generated captions for each image. The best and the second best results are highlighted with **bold** and underline, respectively.

| Method | #Sample | B@1 | B@2 | B@3 | B@4 | R | M | C | S |
|---|---|---|---|---|---|---|---|---|---|
| Div-BS [48] | | 0.837 | 0.687 | 0.538 | 0.383 | 0.653 | 0.357 | 1.405 | 0.269 |
| POS [15] | | 0.874 | 0.737 | 0.593 | 0.449 | 0.678 | 0.365 | 1.468 | 0.277 |
| AG-CVAE [50] | | 0.834 | 0.698 | 0.573 | 0.471 | 0.638 | 0.309 | 1.259 | 0.244 |
| Seq-CVAE [3] | 20 | 0.870 | 0.727 | 0.591 | 0.445 | 0.671 | 0.356 | 1.448 | 0.279 |
| COS-CVAE [33] | | 0.903 | 0.771 | 0.640 | 0.500 | 0.706 | 0.387 | 1.624 | 0.295 |
| AoANet-DML (Ours) | | **0.917** | **0.799** | **0.682** | **0.554** | **0.734** | **0.418** | **1.734** | **0.328** |
| Transformer-DML (Ours) | | 0.915 | 0.788 | 0.663 | 0.526 | 0.726 | 0.417 | 1.704 | 0.325 |
| Div-BS [48] | | 0.846 | 0.698 | 0.555 | 0.402 | 0.666 | 0.372 | 1.448 | 0.290 |
| POS [15] | | 0.909 | 0.787 | 0.672 | 0.550 | 0.725 | 0.409 | 1.661 | 0.311 |
| AG-CVAE [50] | | 0.883 | 0.767 | 0.654 | 0.557 | 0.690 | 0.345 | 1.517 | 0.277 |
| Seq-CVAE [3] | 100 | 0.922 | 0.803 | 0.691 | 0.575 | 0.733 | 0.410 | 1.695 | 0.320 |
| LNFMM [32] | | 0.920 | 0.802 | 0.695 | 0.597 | 0.729 | 0.402 | 1.705 | 0.316 |
| COS-CVAE [33] | | 0.942 | 0.842 | 0.739 | 0.633 | 0.770 | 0.450 | 1.893 | 0.339 |
| AoANet-DML (Ours) | | **0.947** | **0.850** | **0.752** | **0.652** | **0.782** | **0.479** | **1.960** | **0.356** |
| Transformer-DML (Ours) | | 0.946 | 0.849 | 0.750 | 0.649 | 0.780 | 0.474 | 1.953 | 0.354 |

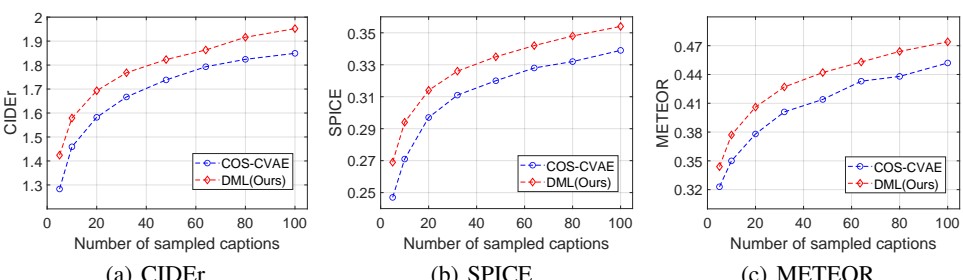

(a) CIDEr                (b) SPICE                (c) METEOR

Figure 4: Comparison between our DML and SoTA baseline COS-CVAE. We show the oracle results of CIDEr, SPICE and METEOR with different numbers of samples.

original Transformer in terms of ClipScore. We further report the best-performed modes *w.r.t.* CIDEr, SPICE and ClipScore for AoANet-DML. In Table 3, the results show that the proposed DML can gain higher scores of CIDEr, SPICE and ClipScore by choosing the suitable mode embedding. These results suggest that our DML not only generates diverse results, but the results of each individual mode also yield high quality.

**Further comparison with COS-CVAE.** To further evaluate the effectiveness of our DML paradigm, we run experiments using the UpDown [2] model (a two-layer LSTM with a visual attention module) for our MIC branch, which is also the same language generation model used by COS-CVAE. The oracle performance of this model is 1.688 and 1.942 in terms of CIDEr for 20 and 100 samples, respectively, which still outperforms the COS-CVAE by a large margin. In fact, UpDown is a strong model that achieves compatible performance with a 6-layer Transformer model in a general image captioning setting (1.099 CIDEr vs. 1.114 CIDEr on Karpathy's test split [21]), which means that two-layer LSTMs may already have enough capacity for the COCO dataset. Moreover, considering that COS-CVAE requires a pre-processing step to construct pseudo supervisions with the help of a pretrained joint vision-language embedding model, the proposed end-to-end learning method could be more convenient to use than COS-CVAE.

### 4.5 Performance Analysis of the CdVAE branch

In this part, we investigate the effect of our mode assignment strategy (Section 3.2) and the fully non-autoregressive (NAT) objective (Section 3.3) used by the CdVAE branch of DML. Specifically, we train Transformer models with a codebook size of $k = 64$ on three different settings: 1) DML w/o NAT objective; 2) DML w/o Hungarian assignment; 3) the proposed DML, and visualize their mode embeddings in the codebook and caption embeddings from the mode encoder output using t-SNE.

Table 3: The performance of image captions generated by different modes. For Transformer-DML, we select some representative modes that exhibit clear language patterns or semantic complexities as shown in Figure 3. For AoANet-DML, we show the best-performed modes in terms of CIDEr, SPICE and ClipScore, respectively (highlighted with underline). The original results of Transformer and AoANet are obtained using the code and settings in [30]. The best results of the two captioning methods on each metric are highlighted with **bold**.

| Models | Mode | B@1 | B@2 | B@3 | B@4 | R | M | C | S | ClipScore |
|---|---|---|---|---|---|---|---|---|---|---|
| Transformer [46] | N/A | 0.751 | 0.589 | **0.448** | 0.338 | 0.558 | 0.276 | 1.119 | 0.207 | 0.990 |
| Transformer-DML | 3 | 0.671 | 0.474 | 0.328 | 0.221 | 0.487 | 0.245 | 0.871 | 0.192 | 1.090 |
| | 7 | 0.677 | 0.497 | 0.359 | 0.253 | 0.499 | 0.247 | 0.958 | 0.189 | 0.967 |
| | 32 | 0.665 | 0.461 | 0.315 | 0.213 | 0.441 | 0.230 | 0.840 | 0.190 | **1.121** |
| | 43 | 0.658 | 0.481 | 0.339 | 0.238 | 0.479 | 0.234 | 0.909 | 0.175 | 1.098 |
| | 58 | **0.752** | **0.590** | **0.448** | **0.340** | **0.559** | **0.277** | **1.124** | **0.208** | 0.945 |
| AoANet [20] | N/A | **0.770** | **0.613** | **0.476** | **0.368** | **0.571** | 0.283 | **1.173** | 0.213 | 0.969 |
| AoANet-DML | 49 | 0.762 | 0.601 | 0.458 | 0.356 | 0.567 | 0.279 | 1.157 | 0.209 | 0.972 |
| | 64 | 0.658 | 0.461 | 0.315 | 0.212 | 0.529 | **0.284** | 0.768 | **0.215** | 1.131 |
| | 44 | 0.747 | 0.583 | 0.440 | 0.339 | 0.557 | 0.279 | 1.123 | 0.210 | **1.195** |

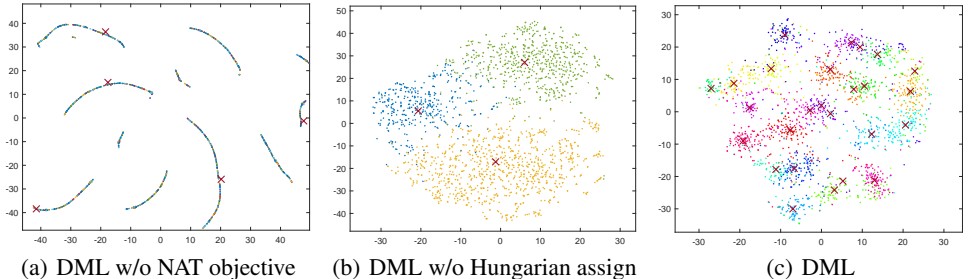

(a) DML w/o NAT objective    (b) DML w/o Hungarian assign    (c) DML

Figure 5: Effects of the Hungarian mode assignment and the fully non-autoregressive decoding objective. We visualize the mode embeddings (red "×") and the caption embeddings ("·" with other colors) on the test set by t-SNE. The inactivated modes are not shown for clarity.

As shown in Figure 5(a), DML w/o NAT only activates 5 of 64 mode embeddings, *i.e.*, all caption embeddings are assigned to the 5 modes. Moreover, the caption embeddings of different modes are mixed together (zoom in to see the details) and are distributed among the twisted spaces nearby the mode embeddings, which means that the mode embeddings and the caption embeddings may not contain useful information. The low oracle CIDEr score (1.207) also verifies this hypothesis.

In Figure 5(b), we show that DML w/o Hungarian assignment activates even fewer modes (*i.e.*, only three), but the caption embeddings now have a more distinct distribution compared with the result in Figure 5(a). This shows the importance of the fully NAT objective in CdVAE. Still, due to the small number of modes, the oracle CIDEr score in this setting is only 1.211, indicating a limited diversity.

Lastly, in Figure 5(c), our DML obtains 29 effective mode embeddings, and the caption embeddings are tightly distributed around their corresponding mode embeddings. Moreover, our model achieves an oracle CIDEr of 1.871. These results demonstrate the effectiveness of the Hungarian mode assignment and the fully non-autoregressive objective for learning distinct and representative mode embeddings.

## 4.6  Ablation Studies on the CdVAE branch

In this section, we provide more analysis on the CdVAE branch of the proposed DML method. Our experiments are based on the Transformer-DML model with a codebook size of 64.

**Asymmetric batch size**    As we mentioned in Section 3.4, the training of the CdVAE branch is much harder than the training of the MIC branch. Thus, we adopt asymmetric batch sizes when training them jointly. Moreover, the gradient from the CdVAE branch will not be back-propagated to the image encoder, so that the whole MIC branch is trained with a batch size $n$ times smaller than that of

Table 4: Oracle results of different batch sizes for CdVAE branch and MIC branch. "#effective modes" indicates the total number of modes that have ever been used during the whole training process. "#sampled caps" refers to how many captions per image are sampled to optimize the MIC branch.

| Batch size | #sampled caps | B1 | B2 | B3 | B4 | R | M | C | S | #effective modes |
|---|---|---|---|---|---|---|---|---|---|---|
| 64 | 1 | 0.932 | 0.826 | 0.719 | 0.608 | 0.761 | 0.454 | 1.871 | 0.342 | 29 |
| 64 | 2 | 0.929 | 0.820 | 0.712 | 0.599 | 0.756 | 0.445 | 1.833 | 0.336 | 28 |
| 64 | 5 | 0.921 | 0.805 | 0.692 | 0.574 | 0.743 | 0.430 | 1.779 | 0.331 | 24 |
| 16 | 2 | 0.917 | 0.799 | 0.683 | 0.561 | 0.737 | 0.427 | 1.760 | 0.327 | 15 |
| 16 | 5 | 0.925 | 0.810 | 0.698 | 0.580 | 0.749 | 0.441 | 1.799 | 0.335 | 17 |

Table 5: Oracle results of different masking strategies for CdVAE branch. "#effective modes" is the total number of modes that have ever been used during the whole training process. "$1.0 \rightarrow 0.0$" means we gradually reduce the masking probability from 1 to 0 throughout the training process.

| Mask probability | B1 | B2 | B3 | B4 | R | M | C | S | #effective modes |
|---|---|---|---|---|---|---|---|---|---|
| 1.0 | 0.932 | 0.826 | 0.719 | 0.608 | 0.761 | 0.454 | 1.871 | 0.342 | 29 |
| 0.5 | 0.833 | 0.675 | 0.513 | 0.342 | 0.633 | 0.337 | 1.384 | 0.262 | 6 |
| $1.0 \rightarrow 0.0$ | 0.879 | 0.740 | 0.604 | 0.448 | 0.685 | 0.377 | 1.579 | 0.294 | 9 |
| 0.0 | 0.790 | 0.619 | 0.439 | 0.270 | 0.590 | 0.301 | 1.207 | 0.228 | 5 |

the CdVAE branch. Here, we show how this strategy benefits the training of our Transformer-DML model. Specifically, during training, we change the number of sampled captions for the MIC branch as well as the total batch size, and the rest settings are the same as in Section 4.2. The results are shown in Table 4.

The first row in Table 4 is the performance of our default setting, which yields the best results among all metrics, and also obtains a larger number of effective modes than other settings. When increasing the number of sampled captions from 1 to 5 for the MIC branch, the performance of Transformer-DML drops clearly. We believe this is due to the over-fitting of the MIC branch caused by the large batch size. When decreasing the total batch size from 64 to 16, we observe a clear reduction in the number of effective modes, which means the CdVAE branch is not sufficiently trained. Moreover, with such a small batch size, the MIC branch will also suffer from under-fitting if the number of sampled captions is small, *i.e.*, oracle CIDEr drops from 1.799 to 1.760 when reducing the number of sampled captions from 5 to 2. These results show the importance of balancing the training of the two branches in the proposed DML.

**Masking strategy of the masked decoder** $\mathcal{D}_m$    When training the CdVAE branch, we use a fully non-autoregressive objective, where the input of the masked decoder $\mathcal{D}_m$ are all `[MASK]` tokens. This prevents the model from using the mode information leaked from the ground-truth tokens, and we find it greatly benefits the learning of modes. Here we evaluate the performance of several different masking strategies, including random masking the input caption using a fixed probability, and random masking with a dynamically changed probability. The results are shown in Table 5.

From the table, the full masking strategy, *i.e.*, a mask probability of 1.0, leads to the best performance, which is also the default setting in DML. Moreover, introducing any additional information to the CdVAE branch, *i.e.*, a mask probability less than 1.0, will severely hamper the learning of modes, where the number of effective modes drops clearly from 29 to less than 10, and the oracle performances also drop significantly.

## 5    Conclusion

In this paper, we study a problem in image captioning that models tend to be biased to generate an "average" caption, which contains the common words or phases only. To tackle this problem, we propose a Discrete Mode Learning (DML) paradigm for image captioning. The idea is to explore multiple rich modes in the training caption corpus to learn a codebook that contains a set of "mode embeddings", which enables the image captioning models to generate different captions based on various modes. Moreover, the proposed DML paradigm can be easily plugged into the existing image captioning models (*e.g.*, Transformer and AoANet) to generate high-quality and diverse captions.

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
