# A  Limitation and Impacts

**Limitation.** Although our method can generate diverse captions based on different mode embeddings in the codebook, the more specific meanings for some of the modes may still be unclear and require more professional analyses.

**Potential negative societal impacts.** We were not able to find some potential negative impact for our image captioning model, but perhaps, the Discrete Mode Learning paradigm may be applied in pre-trained generative models using large-scale web data to learn specific modes for special communities, which may further cause gender/racial bias problems.

# B  Further Discussions

**Beam Search vs. DML.** We evaluate the oracle performance of AoANet and Transformer with beam search (BS) *w.r.t.* both 20 and 100 samples, and provide the results in Table 6. Note that, although beam search achieves remarkable oracle performance, its diversity scores are very low (see Table 1 of the main submission), indicating that most of the sampled captions are very similar. Moreover, beam search is extremely time-consuming: running the Transformer model with beam size 100 takes 10 hours on our machine, while the proposed model only requires $\sim$0.5 hour.

Table 6: Comparison with baselines *w.r.t.* oracle performance (*i.e.*, best-1 quality) on COCO dataset. "#Sample" refers to the number of generated captions for each image.

| Method | #Sample | B@1 | B@2 | B@3 | B@4 | R | M | C | S |
|---|---|---|---|---|---|---|---|---|---|
| AoANet-BS | | 0.912 | 0.808 | 0.700 | 0.580 | 0.746 | 0.458 | 1.829 | 0.323 |
| Transformer-BS | | 0.918 | 0.804 | 0.699 | 0.578 | 0.747 | 0.443 | 1.816 | 0.328 |
| AoANet-DML (Ours) | 20 | 0.917 | 0.799 | 0.682 | 0.554 | 0.734 | 0.418 | 1.734 | 0.328 |
| Transformer-DML (Ours) | | 0.915 | 0.788 | 0.663 | 0.526 | 0.726 | 0.417 | 1.704 | 0.325 |
| AoANet-BS | | 0.955 | 0.883 | 0.805 | 0.71 | 0.817 | 0.543 | 2.068 | 0.368 |
| Transformer-BS | | 0.959 | 0.878 | 0.791 | 0.703 | 0.813 | 0.533 | 2.011 | 0.372 |
| AoANet-DML (Ours) | 100 | 0.947 | 0.850 | 0.752 | 0.652 | 0.782 | 0.479 | 1.960 | 0.356 |
| Transformer-DML (Ours) | | 0.946 | 0.849 | 0.750 | 0.649 | 0.780 | 0.474 | 1.953 | 0.354 |

**Analyses on the evaluation of diversity.** Although the improvement of our method on diversity metrics may not be significant, we would like to argue that:

1) The proposed method focuses on not only the diversity of captions, but more importantly, the controllability. Specifically, we have discovered several distinct and representative language patterns from the generated captions which can be controlled by some of the learned modes. This kind of controllability is lacking in the previous SoTA COS-CVAE as well as other diverse image captioning models.

2) When evaluating the diversity score on the testing split, the nearest neighbor search in the consensus reranking step is performed in the embedding space of VGGNet pretrained on the ImageNet dataset, which may lead to inaccurate search results due to the low representation ability of VGGNet and the domain shift between ImageNet and COCO. Thus, the captions used for diversity evaluation may not be semantically aligned with the test image. Moreover, the diversity metrics themselves only focus on n-gram diversities and ignore the semantic alignment (*i.e.*, they can be easily hacked with random tokens), further exposing the misalignment problem.

Thus, it would be better to consider the results from both Table 1 and Table 2 when assessing the ability of a diverse image captioning model. *I.e.*, a high diversity score and a high oracle score mean that the generated captions may have a high recall rate for the diverse modes that appear in the reference captions, and with each mode the model is able to produce captions with good quality (*i.e.*, fluency and semantic alignment), indicating a human-like behavior. On the other hand, if the diversity score is high but the oracle score is low, this means the generated captions may contain incorrect or misaligned tokens which may not be as diverse as the diversity score indicates.

In this sense, compared with COS-CVAE, our model achieves better performance in terms of the diversity scores and is also less affected by the misalignment problem due to the significantly better oracle scores, suggesting that the real diversity gap between the proposed method and COS-CVAE could be underestimated.

## C   Captions Generated by All Modes

We provide more samples of the generated captions of our method in Figures 6-8. The results are obtained from the Transformer-DML model trained on MSCOCO dataset. The codebook size is 64. The number of effective modes in this model, *i.e.*, the total number of modes that have ever been used during the whole training process, is 29. From the figures, the learned modes exhibit clear and distinct language patterns. For example, almost all captions in mode-2 use the present progressive tense or passive voice; mode-6 tends to use exaggerated words when describing the image, *e.g.*, *"very cute ", "pretty"*; mode-7 tends to follow the pattern *"There is ..."* while mode-43 would like the pattern *"An image of/A close up of ..."*; mode-52 tends to describe the colors in the image; mode-57 uses "and" to combine multiple short phrases for most of the cases; the captions generated by mode-3 are usually very long and semantically rich, while on the contrary, the captions generated by mode-58 are always short and simple. For convenience, we highlight these patterns in the figures.

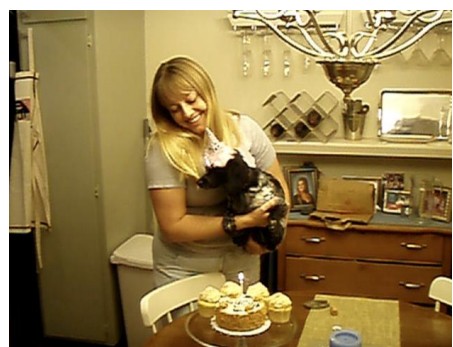

Mode-02: a woman is standing in front of a table with cakes on it
Mode-03: a young woman holding a dog standing in front of a table filled with cakes and cupcakes
Mode-04: a woman is holding a dog and standing in front of a table with cakes on it
Mode-05: a woman holding a dog in her arms in a kitchen
Mode-06: a pretty young lady standing in front of a counter with cakes on it
Mode-07: there is a woman that is standing at a table
Mode-08: a woman is standing in a kitchen with cakes and cup cakes
Mode-10: a woman holding a dog standing in a kitchen
Mode-11: a woman standing in a kitchen holding a dog in her hand
Mode-12: a woman standing at a kitchen counter with cakes and cupcakes
Mode-16: a woman is holding a dog in her hand while standing in front of a table filled with cakes
Mode-17: a woman in a kitchen holding a dog and standing in front of a table full of cakes
Mode-20: a young lady with golden hair standing in front of a table with cakes and cup cakes
Mode-25: a woman standing in a kitchen with a dog and some cakes
Mode-26: a woman at a kitchen counter with cakes and cupcakes
Mode-27: a woman is standing in front of a table with cakes on it
Mode-28: a woman is standing at a kitchen counter with some cakes on the counter
Mode-32: woman with a black dog in her arms at a counter
Mode-33: a woman standing in front of a table with cakes on it
Mode-41: a woman holding a dog while standing in front of a table filled with cakes
Mode-43: an image of a woman that is in the kitchen
Mode-44: a woman is holding a dog in her arms
Mode-52: a woman in a white shirt standing in front of a table with cakes and cupcakes
Mode-54: a woman is holding her dog while she is standing in front of a table full of cakes
Mode-57: a woman in a white shirt a table and some cakes
Mode-58: a woman that is standing in front of a table
Mode-60: a woman smiles while standing in front of a table with cakes on it
Mode-62: a woman standing at a kitchen counter with cakes and cupcakes
Mode-63: a woman holding a dog standing next to a table filled with cakes

Figure 6: Sample-1. Modes marked in red mean the corresponding generated captions have certain and distinct language patterns.

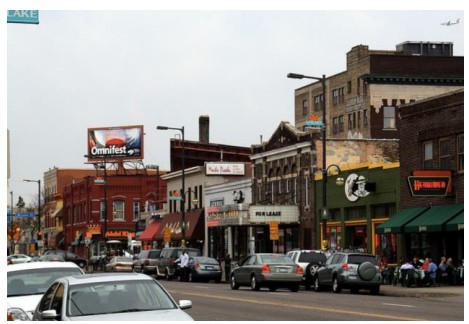

Mode-02: a billboard is shown on the side of a busy street
Mode-03: a city street with a lot of cars parked on the side of the road and people on the sidewalk
Mode-04: a city street with a billboard for a hotel
Mode-05: a street with a bunch of cars parked on the side
Mode-06: a crowded street with a bunch of cars parked on the side of it
Mode-07: there is a very large sign over a busy street
Mode-08: a city street filled with lots of traffic
Mode-10: a street with cars parked along side of it
Mode-11: a city street filled with lots of cars and people
Mode-12: a billboard next to a busy city street
Mode-16: a city street with a billboard for a restaurant
Mode-17: a street with cars parked along the side of the buildings
Mode-20: a group of cars parked in front of a building
Mode-25: a city street filled with various cars
Mode-26: a busy city street with a large billboard and lots of people
Mode-27: a busy street with a lot of people and cars
Mode-28: a large billboard is on a busy street
Mode-32: people and cars parked on the side of the road
Mode-33: a parking lot next to a building with cars parked on the side of it
Mode-41: people are sitting at tables and cars are parked on the side of the street
Mode-43: a view of a city street with a billboard and line of restaurants
Mode-44: a city street with a bunch of cars parked on the side
Mode-52: a busy street with a billboard on the top of a red building
Mode-54: a busy city street with a billboard for a restaurant
Mode-57: a busy street with a bunch of cars parked on the side and people on the sidewalk
Mode-58: a bunch of cars park on the side of a street
Mode-60: a city street is crowded with people and cars
Mode-62: a billboard sitting on the side of a road filled with traffic
Mode-63: a large billboard is sitting on the top of a building

Figure 7: Sample-2. Modes marked in red mean the corresponding generated captions have certain and distinct language patterns.

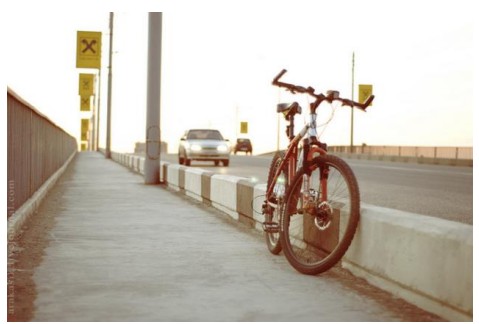

Mode-02: a bike is shown sitting on the side of the road
Mode-03: a red bike with a long handlebars is parked on the side of the road
Mode-04: a red bike with a long handle on a street
Mode-05: a bicycle with a long handlebars is parked on the side of the road
Mode-06: a very nice bike parked by a road
Mode-07: there is a bike that is leaning against a wall
Mode-08: a bike parked on the side of a road
Mode-10: a bike with a long handlebars sitting on a sidewalk
Mode-11: a bike is shown on the side of the road
Mode-12: a bike parked on a sidewalk next to a street
Mode-16: a bike parked on a sidewalk with cars passing by
Mode-17: a red and black bicycle is on a curb
Mode-20: a red bike parked on top of a sidewalk
Mode-25: a bike is sitting on the side of the road
Mode-26: a bike that is leaning against a wall
Mode-27: a bike that is leaning against a curb
Mode-28: a red bike parked on a sidewalk next to a street
Mode-32: red bicycle on side of road with traffic in background
Mode-33: a bike parked next to a wall near a street
Mode-41: a bike is chained up on a concrete wall
Mode-43: a close up of a bike on a city street
Mode-44: a bike is laying on its side on the sidewalk
Mode-52: a red and black bicycle is parked on a sidewalk
Mode-54: a bike that is red and white is on the street
Mode-57: a red and black bicycle some cars and a bridge
Mode-58: red bike is parked next to a curb
Mode-60: a bike is lying on its side on the side of the road
Mode-62: a bike parked on the side walk near a road
Mode-63: a red bike is parked on the side of the road

Figure 8: Sample-3. Modes marked in red mean the corresponding generated captions have certain and distinct language patterns.