# OpenReview forum: "Learning Distinct and Representative Modes for Image Captioning"
_NeurIPS.cc/2022/Conference — NeurIPS 2022 Accept_

### Official Review · Reviewer_g77u · 2022-06-28

**Rating:** 7
**Confidence:** 4
**Soundness:** 4 excellent
**Presentation:** 4 excellent
**Contribution:** 4 excellent

**Summary:**

The authors consider the mode collapse problem in image caption
generation.  They give a diverse caption generation model that differs
from prior work because 1) it's more controllable than conditioning on
randomly-generated latent codes; and 2) the control codes are learned
via VAE clustering, rather than pre-specified by styles in the
training data, e.g., "funny". Several modeling innovations are key to
the end-to-end training working, e.g., a non-autoregressive component,
and a hungarian method for mode assignment. The authors evaluate their
model on the MSCOCO benchmark, demonstrating that their model 1)
generates diverse and controllable-in-style captions based on
interpretable clusters; and 2) performs quite well in the
oracle-conditioned setup according to standard evaluation metrics.

**Questions:**

- Is mode collapse really occurring with image captioning models, or are the decoding methods the reason why the generated captions seem generic?

**Limitations:**

I was a bit disappointed by the discussion of potential negative
societal impacts: what does image captioning have to do with
financial scams? I hope the authors can think a bit more about
automatic image captioning's potential negative impacts, though
I do not think this particular work poses much marginal dual use
risk.

**Strengths And Weaknesses:**

Strengths:

I really liked this work! The introduction frames the problem of mode
collapse quite effectively, and the story flows well. The modeling
components are well-motivated, interesting, and clever. The empirical
results are strong. The ablations are appropriate. The qualitative
results are convincing. I expect that this approach could be useful
for tasks beyond image captioning, too, e.g., in cases where there
isn't one objectively correct generation target, a similar VAE-style
clustering could come in handy, without additional labeling, to give
control over generations when the targets implicitly contain different
styles.

Weaknesses:

- This may be explored in prior work, but I was hoping for some
  information about whether or not the observed mode collapse of
  models is actually P(y|x) mode collapse, or if it is simply a result
  of the decoding algorithms favoring blander outputs. Is it possible
  to score the ground truth captions y_i to see if they collapse to
  the most generic one vs. just relying on the sampled captions, which
  may be affected by the decoding approach?

- While the ablations clearly demonstrate that the non-autoregressive
  training objective (/model) works to spread out mode assignments, I
  wasn't entirely clear how the autoregressive model avoids mode
  collapse at inference time, given the discussion/experiments about
  how "predict the next token" by itself does indeed lead to mode
  collapse. I'm not sure how the authors could address this beyond the
  ablation they already ran, but perhaps, e.g., the attention of the
  transformer based model weighs the VAE cluster token more highly
  after NAT training? Could be interesting to look at.

- It would have been nice to see this model run on datasets other than
  MSCOCO. I wonder if this approach could still work even if there
  were only one reference at training time?


Overall: I really liked this work, and don't have many improvements to
suggest beyond discussing the detail about mode collapse vs. decoding
as discussed above. It's possible that I missed something because I am
less up-to-date on the diverse captioning literature, but if I didn't,
I hope this work appears at NeurIPS.

---

> ### Author Response · Authors · 2022-08-02
> **Authors' Response (Part I)**
>
> Thanks for your valuable comments and questions, our responses are in the following. Hope they can address your concerns.
>
> **Q1: This may be explored in prior work, but I was hoping for some information about whether or not the observed mode collapse of models is actually P(y|x) mode collapse, or if it is simply a result of the decoding algorithms favoring blander outputs. Is it possible to score the ground truth captions y_i to see if they collapse to the most generic one vs. just relying on the sampled captions, which may be affected by the decoding approach? (Is mode collapse really occurring with image captioning models, or are the decoding methods the reason why the generated captions seem generic?)**
>
> **A1**: We try a non-autoregressive decoding algorithm termed Mask-Predict [a] for Transformer and find that the mode collapse problem still exists. For example, about 1.2% captions in the training set of MSCOCO start with “a/an image/picture/view/close up of”, however, in the captions generated by Mask-Predict, this only accounts for 0.5%; about 0.7% training captions start with “there is/are”, and about 3.7% training captions are longer than 15 words, but for captions generated by Mask-Predict these numbers are all 0.0%. Thus, we hypothesize that generative models tend to concentrate on the dominant patterns and suppress the rare patterns in the long-tail target distribution, no matter which decoding algorithm we use.
>
> [a] Ghazvininejad, Marjan, et al. "Mask-Predict: Parallel Decoding of Conditional Masked Language Models." Proceedings of the 2019 Conference on Empirical Methods in Natural Language Processing and the 9th International Joint Conference on Natural Language Processing (EMNLP-IJCNLP). 2019.
>
> **Q2: While the ablations clearly demonstrate that the non-autoregressive training objective /model works to spread out mode assignments, I wasn't entirely clear how the autoregressive model avoids mode collapse at inference time, given the discussion/experiments about how "predict the next token" by itself does indeed lead to mode collapse. I'm not sure how the authors could address this beyond the ablation they already ran, but perhaps, e.g., the attention of the transformer-based model weighs the VAE cluster token more highly after NAT training? Could be interesting to look at.**
>
> **A2**: In the MIC branch, the mode embedding is added to every input word embedding (just like how the positional embedding is added to the word embedding), thus we cannot directly visualize how the transformer decoder utilizes the mode information. Nevertheless, we find that the decoding algorithm is not the key to cause/avoid the mode collapse problem, i.e., whether using autoregressive decoding or non-autoregressive decoding, the mode collapse problem is likely to happen if no mode information is input into the model (see the answer to Q1). In fact, our purpose of using a non-autoregressive decoding algorithm for the CdVAE branch is not to avoid the mode collapse problem, but instead to improve the quality of the learned mode representations. We believe what really alleviates the mode collapse problem is the use of a set of well-learned mode representations (like those in the Figure 5c of the main submission), which can drive the output distribution towards a specific target for both autoregressive decoding and non-autoregressive decoding. Similar observations have also been found in [b].
>
> [b] Deng, Chaorui, et al. "Length-controllable image captioning." European Conference on Computer Vision. Springer, Cham, 2020.
>
> **Q3: It would have been nice to see this model run on datasets other than MSCOCO. I wonder if this approach could still work even if there were only one reference at training time?**
>
> **A3**: Thanks for your suggestion. We run an experiment on a subsampled version of MSCOCO, where each image is paired with only one caption. We find that with careful tuning of the learning rates and batch sizes of the CdVAE branch and the MIC branch, the proposed method is still able to learn representative modes from the training corpus, e.g., we still find some modes tend to use “there is/are”, “a/an view/image of” and color words. Actually, these are the easiest modes to be discovered. With more tuning, we may be able to find more meaningful modes. We have planned to do this in further work, and also planned to apply the proposed method on large-scale image-text datasets like Conceptual Captions, to see if we can learn more useful modes after large-scale pretraining.

---

> > ### Author Response · Authors · 2022-08-02
> > **Authors' Response (Part II)**
> >
> > **Q4: I was a bit disappointed by the discussion of potential negative societal impacts: what does image captioning have to do with financial scams? I hope the authors can think a bit more about automatic image captioning's potential negative impacts, though I do not think this particular work poses much marginal dual use risk.**
> >
> > **A4**: Thank you very much for pointing out this. We were not able to find some potential negative impact for our image captioning model, but perhaps, the Discrete Mode Learning paradigm may be applied in pretrained generative models using large-scale web data to learn specific modes for special communities, which may further cause gender/racial bias problems.

---

> > > ### Comment · Reviewer_g77u · 2022-08-08
> > > **Thanks!**
> > >
> > > Hi Authors!
> > >
> > > Just commenting to let you know that I have read + thought about your responses. I am still quite positive about your work, and appreciate the extra experiment you ran with mask-predict. While it doesn't entirely address my thought that perhaps mode collapse could be, it does bolster your (already strong) case that, at least for these particular decoding algorithms, your new method can be helpful (which is a big step!).

---

> > > > ### Author Response · Authors · 2022-08-09
> > > > **Thanks!**
> > > >
> > > > Thank you very much for your encouragement and valuable comments on our work! We will keep working on the mode collapse problem to get a better understanding.

---

### Official Review · Reviewer_rUVQ · 2022-07-10

**Rating:** 7
**Confidence:** 4
**Soundness:** 3 good
**Presentation:** 3 good
**Contribution:** 3 good

**Summary:**

This paper presents a strategy to generate a diverse set of captions for image captioning. The proposed method is composed of two parts: (1) CdVAE which learns a discrete code book of mode embedding, and (2) MIC which leans to generate captions conditioned on a mode embedding. The proposed method demonstrates slightly better diversity scores and quality scores compared to previous SoTA COS-CVAE.

**Questions:**

- The biggest concern the reviewer has is how to fairly compare with SoTA COS-CVAE method as described in the weakness section. It would be really helpful if the author can either: (1) incorporate their method to the same model architecture as COS-CVAE (LSTM), and/or (2) implement COS-CVAE with a transformer model. In either way, we can rule out the possibility of architectural improvement and focus on the contribution of the proposed method.

- Further analyses of the collapsed mode in Fig.5 as described in the weakness section.

- Some ablations are missing such as K and the loss terms.

**Limitations:**

- Recent VL works typically pre-train a cross-modal transformer model on large image and text corpus and then fine-tune to the downstream target task such as image captioning. It is unknown how the proposed method can be incorporated into the transformer pre-training methods.

- In real-world applications, the users usually only need one caption with the best quality. How does the proposed method search and find the best performing caption among all the modes?

**Strengths And Weaknesses:**

### Strength

1. The training objectives of CdVAE is very similar to [1], but the reviewer considers the incorporation of an existing method for solving an interesting question in a new setting (eg. image captioning) as a contribution. Nevertheless, it would still be good to describe the difference.

1. The proposed method seems to outperform SoTA COS-CVAE in both diversity and quality (more in the weakness section).

1. The core idea and the proposed method is easy to understand and follow.

1. Fig.5 successfully demonstrates how the proposed training techniques affect the learned codebook of mode embedding.

[1] Van Den Oord, Aaron, and Oriol Vinyals. "Neural discrete representation learning." Advances in neural information processing systems 30 (2017).

### Weakness

1. The comparison against SoTA COS-CVAE is unfair. First of all, the performance improvement in terms of the diversity scores in Tab.1 is not significant. The SelfCIDEr score is also missing. Since COS-CVAE released its code, it would be good to re-run the method and include the number. Secondly, COS-CVAE uses LSTM as the language generation model, which is weaker than the transformer model. Therefore, given the slight performance improvement and the usage of a stronger transformer model in this work, it is hard to conclude that the performance improvement compared to SoTA COS-CVAE is actually coming from the proposed method.

1. In Sec. 4.5, only qualitative results and the oracle CIDEr scores are provided. What about the diversity scores? Furthermore, any discussions or analyses of the collapsed modes? Do they really lead to the same output samples? Even the proposed method has a mode collapse issue. Please provide further studies.

1. In line 208, it says the model embedding is added to the word embedding of each token in $y_i$. Does that mean for each $w_{j<t}$ in Eq.5 when generating the next token $w_t$?

1. In Tab.2, baseline methods can also generate multiple samples by beam search. It would be helpful to include those results for AoANet and Transformer as a baseline comparison.

1. K (total number of modes in the codebook) seems to be an important hyper-parameter. How does the performance vary with respect to K?

1. The ablation study of the loss terms (1) $| \text{sg}[e(y_i)] − q(y_i)|^2$ and (2) $| e(y_i) − \text{sg}[q(y_i)]|^2$ in Eq.2 is not provided.

---

> ### Author Response · Authors · 2022-08-02
> **Authors' Response (Part I)**
>
> Thanks for your valuable comments, our responses are in the following. Hope them can address your concerns.
>
> **Q1: The training objectives of CdVAE are very similar to [a], but the reviewer considers the incorporation of an existing method for solving an interesting question in a new setting as a contribution. Nevertheless, it would still be good to describe the difference.**
>
> [a] Van Den Oord, Aaron, and Oriol Vinyals. "Neural discrete representation learning." Advances in neural information processing systems 30 (2017).
>
> **A1**: While the training objective is indeed inspired by [a], the resultant model is totally different. Specifically, we decouple the input (caption) of CdVAE into two types of information: the mode and the content, where the encoder of CdVAE only extracts the mode information from the caption, and for the content information, we adopt the image representations shared from the MIC branch. This enables the CdVAE branch to explicitly focus on the learning of mode. On the other hand, the original method in [a], as well as previous Conditional VAE-based diverse image captioning methods (like COS-CVAE), do not have this information decoupling procedure.
> More importantly, as we have mentioned in the main submission, simply applying the learning method of [a] to our setting did not work well (see Figure 5 of the main submission). To aid this, we introduce several new components, i.e., the Hungarian mode assignment and the fully non-autoregressive objective, which have been proved to be the key ingredients to make the proposed method work. Lastly, we also make several important designs to make the convergence speed of the two branches compatible and thereby prevent overfitting, as described in Section 3.4 of our main submission and verified in Section A of the supplementary materials. We will clarify these differences carefully in the revision.
>
> **Q2: The performance improvement in terms of the diversity scores in Table 1 is not significant. The SelfCIDEr score is also missing.**
>
> **A2**: We evaluate the SelfCIDEr of COS-CVAE which is 0.79, inferior to the performance of our method (0.83). Although the improvement of our method on diversity metrics may not be significant, we would like to argue that:
> 1) The proposed method focuses on not only the diversity of captions, but more importantly, the controllability. Specifically, we have discovered several distinct and representative language patterns from the generated captions which can be controlled by some of the learned modes (see the visualization results in the main submission and the supplementary material). This kind of controllability is lacking in the previous SoTA COS-CVAE as well as other diverse image captioning models.
> 2) When evaluating the diversity score on the testing split, the nearest neighbor search in the consensus reranking step is performed in the embedding space of VGGNet pretrained on the ImageNet dataset, which may lead to inaccurate search results due to the low representation ability of VGGNet and the domain shift between ImageNet and COCO. Thus, the captions used for diversity evaluation may not be semantically aligned with the test image. Moreover, the diversity metrics themselves only focus on n-gram diversities and ignore the semantic alignment (i.e., they can be easily hacked with random tokens), further exposing the misalignment problem.
>
> Thus, it would be better to consider the results from both Table 1 and Table 2 when assessing the ability of a diverse image captioning model. I.e., a high diversity score and a high oracle score mean that the generated captions may have a high recall rate for the diverse modes that appear in the reference captions, and with each mode the model is able to produce captions with good quality (i.e., fluency and semantic alignment), indicating a human-like behavior. On the other hand, if the diversity score is high but the oracle score is low, this means the generated captions may contain incorrect or misaligned tokens which may not be as diverse as the diversity score indicates.
>
> In this sense, compared with COS-CVAE, our model achieves better performance in terms of the diversity scores and is also less affected by the misalignment problem due to the significantly better oracle scores, suggesting that the real diversity gap between the proposed method and COS-CVAE could be underestimated. We will give more discussions on this in the revision.

---

> > ### Author Response · Authors · 2022-08-02
> > **Authors' Response (Part II)**
> >
> > **Q3: COS-CVAE uses LSTM as the language generation model, which is weaker than the transformer model. Therefore, it would be really helpful if the author can either: (1) incorporate their method to the same model architecture as COS-CVAE (LSTM), and/or (2) implement COS-CVAE with a transformer model.**
> >
> > **A3**: We run experiments using the UpDown [b] model (a two-layer LSTM with a visual attention module) for our MIC branch, which is also the language generation model used by COS-CVAE. The oracle performance of this model is 1.688 and 1.942 in terms of CIDEr for 20 and 100 samples, respectively, still outperforms the COS-CVAE by a large margin. In fact, UpDown is a strong model that achieves compatible performance with a 6-layer Transformer model in a general image captioning setting (1.099 CIDEr vs. 1.114 CIDEr on Karpathy’s test split), which means that two-layer LSTMs may already have enough capacity for the COCO dataset. We will give more discussions on this in the revision.
> > Moreover, considering that COS-CVAE requires a pre-processing step to construct pseudo supervisions with the help of a pretrained joint vision-language embedding model, the proposed end-to-end learning method could be more convenient to use than COS-CVAE.
> >
> > [b] Anderson, Peter, et al. "Bottom-up and top-down attention for image captioning and visual question answering." Proceedings of the IEEE conference on computer vision and pattern recognition. 2018.
> >
> > **Q4: In Sec. 4.5, only qualitative results and the oracle CIDEr scores are provided. What about the diversity scores?**
> >
> > **A4**: We cannot directly compute the diversity scores under a fair setting for the models in Figure 5a (DML w/o NAT) and Figure 5b (DML w/o Hungarian assign) since they only have five and three effective modes respectively and cannot provide enough candidates for consensus reranking. Nevertheless, we still calculate the SelfCIDEr scores for the models in Figure 5 by skipping the consensus reranking step and calculating the score within three randomly sampled captions for each image. The diversity scores are 0.64, 0.73, and 0.86 for DML w/o NAT, DML w/o Hungarian assign, and the original DML.
> >
> > **Q5: In Sec. 4.5, any discussions or analyses of the collapsed modes? Do they really lead to the same output samples? Even the proposed method has a mode collapse issue.**
> >
> > **A5**: In Section 4.5, we train three models with a default codebook size of 64. The first two models, DML w/o NAT and DML w/o Hungarian assign only activate a few entries of the codebook (five and three, respectively), and the output samples generated by different modes are indeed very similar for both of these two models, indicating a severe mode collapse issue. This is also reflected by their low diversity scores (see **A4**). The proposed DML activates 29 out of 64 entries and the output samples are very diverse and have some clear language patterns (see the diversity scores in **A4** and the visualization results in the supplementary material). Although it does not fully utilize the codebook, we hypothesize that the distinct and representative modes contained in the training corpus of the COCO dataset may not be large since it only contains descriptive sentences. Thus, the proposed DML can effectively alleviate the mode collapse issue. We will give more discussions on the collapsed mode in the revision.
> >
> > **Q6: In line 208, it says the mode embedding is added to the word embedding of each token in $y_i$. Does that mean for each $w_{j<t}$ in Eq.5 when generating the next token $w_t$?**
> >
> > **A6**: Yes, we add the mode embedding to previous word tokens $w_{j<t}$ element-wisely when generating the next token $w_t$. We will make it clear in the revision.

---

> > > ### Author Response · Authors · 2022-08-02
> > > **Authors' Response (Part III)**
> > >
> > > **Q7: In Tab.2, baseline methods can also generate multiple samples by beam search. It would be helpful to include those results for AoANet and Transformer as a baseline comparison.**
> > >
> > > **A7**: We evaluate the oracle performance of AoANet and Transformer with beam search (BS) w.r.t. both 20 and 100 samples, and provide the results in the table below.
> > >
> > > |Method | #Sample | B@1 | B@2 | B@3 | B@4 | R | M | C | S |
> > > | ------------ | :------------: | :------------: | :------------: | :------------: | :------------: | :------------: | :------------: | :------------: | :------------: |
> > > | AoANet-BS | 20 | 0.912 | 0.808 | 0.700 | 0.580 | 0.746 | 0.458 | 1.829 | 0.323 |
> > > | Transformer-BS | 20 | 0.918 | 0.804 | 0.699 | 0.578 | 0.747 | 0.443 | 1.816 | 0.328 |
> > > |AoANet-DML | 20 |0.917| 0.799 |0.682| 0.554| 0.734| 0.418 |1.734| 0.328|
> > > |Transformer-DML | 20 | 0.915| 0.788| 0.663| 0.526| 0.726| 0.417 |1.704| 0.325|
> > > | AoANet-BS | 100 | 0.955 | 0.883 | 0.805 | 0.710 | 0.817 | 0.543 | 2.068 | 0.368 |
> > > | Transformer-BS | 100 | 0.959 | 0.878 | 0.791 | 0.703 | 0.813 | 0.533 | 2.011 | 0.372 |
> > > |AoANet-DML | 100 |0.947| 0.850| 0.752| 0.652| 0.782| 0.479| 1.960| 0.356|
> > > |Transformer-DML| 100 |0.946| 0.849| 0.750 |0.649 |0.780| 0.474| 1.953| 0.354|
> > >
> > > Note that, although beam search achieves remarkable oracle performance, its diversity scores are very low (see Table1 of the main submission), indicating that most of the sampled captions are very similar. Moreover, beam search is extremely time-consuming: running the Transformer model with beam size 100 takes 10 hours on our machine. While the proposed model only requires 20 minutes to generate 100 captions. We will include the results and the discussions in the revision.
> > >
> > > **Q8: K (total number of modes in the codebook) seems to be an important hyper-parameter. How does the performance vary with respect to K?**
> > >
> > > **A8**: In our main submission, we have trained Transformer-DML and AoANet-DML with different codebook sizes (i.e., k = 20, 64, and 100, which can be found in Table 2 and Section 4.5), and evaluated the oracle results. Specifically, Transformer-DML achieves 1.704, 1.871 and 1.953 oracle CIDEr scores with k = 20, 64, and 100, respectively. Moreover, the numbers of effective modes for k = 20, 64, and 100 are 20, 29, and 34, respectively. A similar trend is also observed in the results of AoANet-DML, showing that the oracle performance improves with a larger codebook size, which is reasonable since more effective modes are learned so the recall rate for the modes that appear in the testing split is increased.
> > >
> > > **Q9: The ablation study of the second and the last loss terms in Eq.2 is not provided.**
> > >
> > > **A9**: As we have mentioned in Section 3.1 of the main submission, we design the loss function following [a] to optimize our mode encoder, masked decoder, and the codebook. Since the codebook will only be optimized by the second loss term, if we remove it, the mode embeddings will never be updated and the whole method will not work. For the last term, we have already tried different values for $\beta$ and found the performance is quite robust. Similar observations have also been found in [a]. Thus, we directly adopt the default value of $\beta$ in [a].
> > >
> > > [a] Van Den Oord, Aaron, and Oriol Vinyals. "Neural discrete representation learning." Advances in neural information processing systems 30 (2017).
> > >
> > > **Q10: Recent VL works typically pre-train a cross-modal transformer model on large image and text corpus and then fine-tune to the downstream target task such as image captioning. It is unknown how the proposed method can be incorporated into the transformer pre-training methods.**
> > >
> > > **A10**: It is an interesting topic to combine the proposed method with large-scale vision-language pretraining. Considering that it may need a lot of effort to tune, we plan to explore this direction in further work. We have several initial ideas. The simplest one is to initialize the MIC branch with a pretrained vision-language model, which may lead to higher performance for each learned mode. Besides, we can also directly pretrain the proposed method on large-scale image-text data with the same training objective used in this submission to facilitate a mode-content disentangled vision-language pretraining.

---

> > > > ### Author Response · Authors · 2022-08-02
> > > > **Authors' Response (Part IV)**
> > > >
> > > > **Q11: In real-world applications, the users usually only need one caption with the best quality. How does the proposed method search and find the best performing caption among all the modes?**
> > > >
> > > > **A11**: We would like to answer this question from two aspects. On the one hand, it is usually hard to define the “best performing caption” in real-world scenarios. Different users would prefer different caption styles. In this sense, it is better to provide them with multiple captions that have distinct and representative modes. Some advertisement/design text generation applications have indeed adopted this strategy. On the other hand, if the “best performing caption” does exist, we can still train a “best mode selector” to predict the “most suitable mode” directly from the image through policy gradient techniques. This should be effective for some specific modes, like if an image is semantically complicated, a long and detailed caption could be more suitable than a short and brief caption, and if an image is semantically simple, a brief caption is generally preferred. We leave this to future work.

---

> ### Comment · Reviewer_rUVQ · 2022-08-05
> **Score adjustment**
>
> The reviewer has gone through the rebuttal provided by the author. The authors have successfully addressed the reviewer's concern and provided convincing explanations, especially in the unfair comparison part in terms of metrics and model architecture. Therefore, the reviewer decided to raise the score and suggest accepting this paper.

---

> > ### Author Response · Authors · 2022-08-09
> > **Thanks!**
> >
> > Thank you for the insightful comments on our work! We will include the discussions and explanations of the comparison part w.r.t. the metrics and mode architecture in our revised paper and make them clearer.

---

### Official Review · Reviewer_eCow · 2022-07-11

**Rating:** 6
**Confidence:** 5
**Soundness:** 3 good
**Presentation:** 3 good
**Contribution:** 2 fair

**Summary:**

This paper proposed a diverse image captioning model, namely conditioned discrete variational autoencoder (CdVAE), which is able to learn multiple modes of captions. Plus, to learn better mode embeddings, a masked decoder and Hungarian algorithm are introduced into the training process. Finally, extensive experiments are conducted, showing that the proposed model can generate diverse captions for an input image.

**Questions:**

The CdVAE branch is separate from the image captioning branch, and CdVAE can be trained using a large-scale dataset composed of texts, so I am wondering about the performance of using a large-scale dataset.

**Ethics Review Area:**

["I don’t know"]

**Strengths And Weaknesses:**

The weakest part of the paper is that the framework seems just combines VQ-VAE and a normal image captioning model, though some findings are interesting, such as directly using VQ-VAE cannot capture many modes and to mitigate this issue, this paper introduces Hungarian algorithm and masked decoder. Another strength of the paper is that extensive experiments are conducted and the performance of the paper is relatively good. Plus, the paper is well written and can be easily followed.

---

> ### Author Response · Authors · 2022-08-02
> **Authors' Response**
>
> **Q1: The weakest part of the paper is that the framework seems just combines VQ-VAE and a normal image captioning model, though some findings are interesting, such as directly using VQ-VAE cannot capture many modes and to mitigate this issue, this paper introduces Hungarian algorithm and masked decoder.**
>
> **A1**: Thanks for your valuable comments. While the proposed method is indeed inspired by VQ-VAE, the resultant model is totally different. Specifically, we decouple the input (caption) of CdVAE into two types of information: the mode and the content, where the encoder of CdVAE only extracts the mode information from the caption, and for the content information, we adopt the image representations shared from the MIC branch. This enables the CdVAE branch to explicitly focus on the learning of mode. On the other hand, the original VQ-VAE, as well as previous Conditional VAE-based diverse image captioning methods, do not have this information decoupling procedure.
> More importantly, simply applying the learning method of VQ-VAE to our setting did not work well as shown in Figure 5 of our main submission. To aid this, we introduce several new components, i.e., the Hungarian mode assignment and the fully non-autoregressive objective, which have been proved to be the key ingredients to make the proposed method work. Lastly, we also make several important designs to make the convergence speed of the two branches compatible and thereby prevent overfitting, as described in Section 3.4 of our main submission and verified in Section A of the supplementary materials. We will clarify these differences carefully in the revision.
>
> **Q2: The CdVAE branch is separate from the image captioning branch, and CdVAE can be trained using a large-scale dataset composed of texts, so I am wondering about the performance of using a large-scale dataset.**
>
> **A2**: Thanks for this suggestion. The proposed CdVAE can be trained using a large-scale text-only dataset as long as there are two inputs: one provides the mode information and the other provides the content information. Since this requires a lot of data and computational resources, as well as careful network architecture tuning, we plan to explore it in further work.

---

> > ### Comment · Reviewer_eCow · 2022-08-08
> > **Reply**
> >
> > I have read the feedback and other reviewers' comments. Basically, I think this paper is interesting since diverse image captioning is important for both research and applications and the proposed model outperforms existing approaches. But I still strongly recommend using a pre-trained CdVAE, which could be helpful.
> >
> > I will raise the rating to weak accept.

---

> > > ### Author Response · Authors · 2022-08-09
> > > **Thanks!**
> > >
> > > Thank you for your valuable suggestions! We agree that it would be helpful for pre-training our CdVAE on a larger-scale dataset. We will consider it as our future work. Also thanks for your interest in our paper.

---

### Official Review · Reviewer_Z9St · 2022-07-11

**Rating:** 5
**Confidence:** 4
**Soundness:** 3 good
**Presentation:** 2 fair
**Contribution:** 1 poor

**Summary:**

This paper aims to learn distinct and representative modes for image captioning. The paper propose Discrete Mode Learning (DML), which is a good method to improve the diversity of image captioning. The DML optimizes a dual architecture that consists of an image-conditioned discrete variational autoencoder (CdVAE) branch and a mode-conditioned image captioning (MIC) branch.


**Questions:**

1. The common captioning models generate one caption for each image, while this work and some previous related works generate 20 or 100 captions for each image. Which kind is better?
2. The visual language pre-training models achieve good performance on image captioning recently. Do you think our new methods should apply to those methods?

**Limitations:**

There is no potential negative societal impact of this work.

**Strengths And Weaknesses:**

Strengths:
1. The experimental results show that the DML works well.
2. The Discrete Mode Learning (DML) method is reasonable, and the training loss is well designed.

Weaknesses:
1. The main models used in this paper are Transformer (published in 2017) and AoANet (published in 2019), which are out of date. M2Transformer and XLAN are open sources. And many pre-training models (like Oscar [3r]) are also open source.
2. Since the codebook is in latent space, it is hard to control the semantic meaning of a certain mode. It is a weak point of the latent space based method. The diversity comes from different modes but we can not know which mode is needed. For example, people rarely have the patience to choose the needed captions from 100 samples.
3. This paper uses the m-RNN split, and the most recent image captioning works use Karpathy split [4r]. The results are hard to compare in this case. It will be a contribution if this paper also turns the previous typical works (e.g. COS-CVAE) into the Karpathy split.
4. Paper writing needs to be refined.
In Figure 2, the text of "y1, y2, y3" appears three times exactly the same, which does not help to show the main idea.


[1r] Marcella Cornia et al. “Meshed-Memory Transformer for Image Captioning” computer vision and pattern recognition (2019).
[2r] Yingwei Pan et al. “X-Linear Attention Networks for Image Captioning” computer vision and pattern recognition (2020).
[3r] Xiujun Li et al. “Oscar: Object-Semantics Aligned Pre-training for Vision-Language Tasks” European conference on computer vision (2020).
[4r] Andrej Karpathy and Li Fei-Fei. “Deep Visual-Semantic Alignments for Generating Image Descriptions” IEEE Transactions on Pattern Analysis and Machine Intelligence (2014).

---

> ### Author Response · Authors · 2022-08-02
> **Authors' Response (Part I)**
>
> Thanks for your comments, our responses are in the following. Hope they can address your concerns.
>
> **Q1: The main models used in this paper are Transformer (2017) and AoANet (2019), which are out of date. Consider more recent models like M2Transformer (2020), XLAN (2020), and pretraining models like Oscar (2020).**
>
> **A1**: The proposed Discrete Mode Learning (DML) is a general learning paradigm and we expect it can be easily applied in many existing image captioning models and improve their controllability and diversity. Therefore, we choose two widely-used and representative architectures, i.e., Transformer and AoANet, to show the generalization ability of DML. Specifically, most current state-of-the-art image captioning models, like M2Transformer and many vision-language pretraining models, are based on the Transformer architecture. By showing the ability of DML on Transformer, we show the potential ability of DML on these Transformer-based state-of-the-art models.
> Moreover, the performance of Transformer and AoANet is also competitive with the state-of-the-art M2Transformer and XLAN. With Self-Critical training and without vision-language pretraining, Transformer achieves 127.7 CIDEr score [a] and AoANet achieves 129.8 [b], while M2Transformer and XLAN achieve 131.2 [c] and 132.0 [d], respectively, indicating a close performance gap. Therefore, we believe Transformer and AoANet are two representative and strong baselines when vision-language pretraining is not performed, and they are good enough to illustrate the superior performance led by our DML model. Thanks for this suggestion, we will also try our method on more powerful pretraining models (which is not the major focus of this paper) in future work.
>
> [a] [https://github.com/ruotianluo/self-critical.pytorch/blob/master/MODEL_ZOO.md](https://github.com/ruotianluo/self-critical.pytorch/blob/master/MODEL_ZOO.md)
>
> [b] Huang, Lun, et al. "Attention on attention for image captioning." Proceedings of the IEEE/CVF international conference on computer vision. 2019.
>
> [c] Cornia, Marcella, et al. "Meshed-memory transformer for image captioning." Proceedings of the IEEE/CVF conference on computer vision and pattern recognition. 2020.
>
> [d] ​​Pan, Yingwei, et al. "X-linear attention networks for image captioning." Proceedings of the IEEE/CVF conference on computer vision and pattern recognition. 2020.
>
> **Q2: Since the codebook is in latent space, it is hard to control the semantic meaning of a certain mode. It is a weak point of the latent space based method. The diversity comes from different modes but we can not know which mode is needed. For example, people rarely have the patience to choose the needed captions from 100 samples.**
>
> **A2**: As we have mentioned in the Limitation section, without explicit annotations for the mode of the captions, it is indeed difficult to control the semantic meaning of each learned mode. However, compared with the continuous latent code used in other latent space based methods, our discrete modes are more interpretable where some of them facilitate clear and specific language patterns, as shown in the visualization results in the main submission and the supplementary material. In this sense, people can acquire captions with their preferred styles by choosing the corresponding modes, which is very useful in scenarios like advertisement/design text generation.
> Moreover, the ability of learning interpretable and representative modes via a purely unsupervised manner also makes the proposed model much more convenient to use compared with those that require additional annotations, and the learned modes are also not restricted within the predefined set. These are the specific advantages of the proposed method.
>
> **Q3: This paper uses the m-RNN split, and the most recent image captioning works use Karpathy split [4r]. The results are hard to compare in this case. It will be a contribution if this paper also turns the previous typical works (e.g. COS-CVAE) into the Karpathy split.**
>
> **A3**: Since our submission mainly focuses on the task of Diverse and Controllable Image Captioning, we follow the previous works in this area to use the m-RNN split of the COCO dataset, which is fair in comparison. When evaluating the performance of each mode, the baseline models (AoANet and Transformer) are also trained and evaluated under the m-RNN split, as we have mentioned in Table 3 of our main submission, which is also fair in comparison.
>
> **Q4: Paper writing needs to be refined. In Figure 2, the text of "y1, y2, y3" appears three times exactly the same, which does not help to show the main idea.**
>
> **A4**: Thanks for your suggestion, we will revise the figure carefully.

---

> > ### Author Response · Authors · 2022-08-02
> > **Authors' Response (Part II)**
> >
> > **Q5: The common captioning models generate one caption for each image, while this work and some previous related works generate 20 or 100 captions for each image. Which kind is better?**
> >
> > **A5**: The general image captioning models usually cannot make a good balance between quality and diversity & controllability. Thus, diverse and controllable image captioning methods are proposed to tackle this problem, which can be as good and efficient as general image captioning models when generating one caption for each image, while also having the ability to produce multiple different captions describing the image in various ways. In this sense, the latter is more close to human behavior (humans can describe an image in various ways) and could be more useful in practice.
> >
> > **Q6: The visual language pre-training models achieve good performance on image captioning recently. Do you think our (your) new methods should apply to those methods?**
> >
> > **A6**: Thanks for the suggestion. The proposed Discrete Mode Learning (DML) is a general learning paradigm and does not rely on specific backbones. This is why we can deploy it on both Transformer and AoANet. Large-scale vision-language pretraining models are normally built based on Transformer structure so we believe our DML can be applied to them as well. However, large-scale vision-language pretraining models generally require huge costs to train. Thus, we have planned to do this in further work.

---

> ### Comment · Reviewer_Z9St · 2022-08-07
> **Score adjustment**
>
> I have read the comments from the authors and other reviewers. I decide to raise the rating to borderline accept since
>
> (1) The authors have addressed most of my concerns.
>
> (2) The other reviewers give quite positive comments for this paper on improving VQ-VAE and applying it to the diverse image captioning task.
>
> (3) I still believe that better baselines (e.g. visual language pre-training models) and Karpathy split could make this work more solid.

---

> > ### Author Response · Authors · 2022-08-09
> > **Thanks!**
> >
> > Thank you again for the constructive comments! We will add the clarifications in the response to our revised paper.

---

### Meta-Review · Area_Chair_kv43 · 2022-08-29

**Recommendation:** Accept
**Confidence:** Certain

**Metareview:**

The paper tackles the problem of mode collapse in image captioning and provide a method for generating diverse captions. The proposed approach uses a VAE to learn various modes, each of which can produce a different caption, along with various technical innovations to train the model. Experiments with two models on MS COCO demonstrate the effectiveness of the approach. The paper offers useful insights for the challenging and important problem of diverse captioning and will inform future work in this space.  I encourage the authors to make the revisions suggested by reviewers to improve the clarity of the writing and also include adequate justification for the choice of their base models.

**Award:**

No

---

### Decision · Program_Chairs · 2022-09-14

Accept